# A Spectral Method for Off-Policy Evaluation in Contextual Bandits under Distribution Shift

## Abstract

Contextual bandits capture the partial-feedback nature in an interactive system.

Algorithms for contextual bandits have wide applications in automated decision making such as recommender system and automated stock trading. Evaluating the cumulative reward of a target policy given the historical trajectories of a logging policy (i.e. off-policy evaluation) in contextual bandit setting is a task of importance, as it provides an estimate of the performance of a new policy without experimenting with it.

One (common and well-studied) solution is the Inverse Propensity Score (IPS) estimator. The idea of such methods is to estimate the expectation through importance sampling (i.e. re-weighting the data with a ratio associated with the logging and evaluation policy). Existing work assumes the stationarity of the distribution over context space, which is not always true in a real-world scenario. More practical modeling considers the shift of context/reward distributions between the logged data and the contexts observed in order to evaluate a target policy in the future.

Such a problem is difficult in general due to the high-Dimensionality of the context space, as observed in our experiments. In this paper, we propose an intent shift model which proposes to introduce an intent variable to capture the distributional shift on context and reward. Under the intent shift model, we propose a consistent spectral estimator for the reweighting factor and its finite-sample analysis and provide an MSE bound on the performance of our final estimator. Experiments show that our estimator outperforms the existing ones. W

[1] Anonymous Institution, Anonymous City, Anonymous Region, Anonymous Country. Correspondence to: Anonymous Author <anon.email@domain.com>.

Preliminary work. Under review by the International Conference on Machine Learning (ICML). Do not distribute.

## 1. Introduction

Automated decision making is an important task in machine learning, encompassing practical applications such as recommendation system, web search and stock trading ((Li et al., 2010; Bottou et al., 2013; Tang et al., 2013; Wang et al., 2014)). These problems are often studied under the contextual bandits model ((Abe et al., 2003; Auer, 2002)) where the agent (the algorithm) chooses an action $a$ from the action set $\mathcal{A}$ on observation of a context $x \in X \subset \mathbb{R}^d$, which is drawn from a fixed unknown distribution, i.e. the a stationary environment. A reward $r$ is then revealed, according to a distribution specified by the context-action pairs. The goal is to find a good policy that maximizes the expected reward, (e.g., click through rate (CTR) in the recommendation setting). In some scenario, the safety (or its opposite, risk) of a decision making system is of core concern. That is, the performance of a new policy need to be assessed before put into use. Off-policy evaluation provides a way to estimate of the performance of policies based on historical data. Such estimation allows a controlled risk for deploying a new policy.

Different estimators for off-policy evaluation, under the assumption that the contexts and rewards are drawn from an unknown stationary distribution/environment, have been proposed for contextual bandits and reinforcement learning settings ((Jiang & Li, 2015; Wang et al., 2017; Thomas et al., 2015; Thomas & Brunskill, 2016)). However for real-world settings, stationary distribution might be too strong an assumption. In recommendation systems for e-commerce, the potential item for purchase for users varies at different time of a year. Apparently one would expect a decrease in the sales of coats during summertime. Works on domain adaption ((Lipton et al., 2018; Azizzadenesheli et al., 2018; Reddi et al., 2015; Gretton et al., 2009)) aim to address such distribution mismatch in supervised learning.

In this paper, we address the problem of off-policy evaluation when there is not only a mismatch between behavior and target policy, but also a distribution shift for the context and reward model. Such distribution shift is hard to address with conventional algorithms such as Kernel Mean Matching(Gretton et al., 2009) due to the high-dimensionality of contexts, as we show in Figure 2a. We address the prob-

lem of context distribution shift by proposing an intent shift model where a hidden intent variable governs the distribution shift. Under the intent shift model, we introduce an off-policy estimator that balances between bias and variance of the estimator and achieves a low mean squared error. We use a spectral method to consistently estimate the distribution shift and provide the sample complexity needed to confidently approximate the true underlying distribution shift as shown in Theorem 3.7. Empirical study on semi-synthetic data illustrates that the our estimator outperform the existing estimators.

### 1.1. Related Work

**Off-policy Evaluation**  (Jiang & Li, 2015)(Wang et al., 2017) Off-policy evaluation is a line of work in contextual bandit and reinforcement learning that considers evaluation of a new policy with data collected by an old policy. This is important in the tasks where deploying the new policy is either expensive or risky. A popular baseline is the inverse propensity score (IPS) method, that reweight the logging data with a reweight factor. (Jiang & Li, 2015) provides an estimator to have both low variance and low bias by combining IPS and model-based estimator. (Wang et al., 2017) provides an adaptive estimator and an minmax lowerbound.

**Bandits under with distribution mismatch**  Another related topic is (contextual) bandit with distribution mismatch. (Zhang et al., 2019) considers the scenario, where the fully-observed (as in supervised learning) historical data are available, yet there is a distribution mismatch in the historical data and the environment that the algorithm can interact with.

**Domain Adaptation**  A closely related topic is the label shift setting in the domain adaptation literature (Lipton et al., 2018), (Azizzadenesheli et al., 2018). This is a distribution shift setting in supervised learning, where the distribution shift in samples is only due the distribution shift on the labels: $p_\mathcal{S}(x|y) = p_\mathcal{T}(x|y)$, and $p_\mathcal{S}(y) \neq p_\mathcal{T}(y)$. The difference of our setting lies in the fact that, in our case, the ground truth labels are not available. The major difference in setting compared with (Lipton et al., 2018) is that, in supervised learning, the ground truth labels are available for source domain data, while in our setting such information is mising for both source and target domain.

**Spectral Method and crowd sourcing**  The similar technique of spectral method has been used in inference in probabilistic models (Hsu & Kakade, 2013)(Anandkumar et al., 2012). (Zhang et al., 2014)(Chaganty & Liang, 2013) provides an application of spectral method on crowdsourcing.

## 2. Setting and notation

In a contextual bandit problem, an agent interact with the environment as follows:

1. At each iteration, an context $x_t \in \mathcal{X}$ is drawn according to an unknown but fixed distribution $\mathcal{D}_\mathcal{X}$ and revealed to the agent.

2. the agent picks an action $a_t \in \mathcal{A}$ given the revealed context $x_t$ according to some internal policy. Throughout this paper we will focus on stationary and stochastic policies (i.e. $a_t \sim \pi(\cdot|x_t)$, where $\pi(\cdot|x)$ is a time-independent distribution over the action space.)

3. an reward is drawn from the a distribution over $[0, R_{\max}]$ specified by the action-context pair: $r_t \sim \mathcal{D}_R(\cdot|x_t, a_t)$. Let $r^*(x, a)$ denote the expected reward given the action-context pair: $r^*(x, a) = \mathbb{E}_{r \sim \mathcal{D}_\mathcal{R}(\cdot|x,a)}[r]$.

We use the shorthand of $(x_t, a_t, r_t) \sim (\mathcal{D}, \pi)$ to denote a triplet sampled with respect to the joint distribution induced by $(\mathcal{D}_\mathcal{X}, \pi, \mathcal{D}_\mathcal{R})$.

### 2.1. Off-policy evaluation under distribution shift

Off-policy evaluation is commonly required in online decision making problems due to the exploration and exploitation tradeoff: The behavior policy $\mu$ is usually suboptimal for the purposes of exploration and different from the target policy $\pi$ the goal of which is to improve cumulative rewards. The goal of off-policy evaluation (OPE) is to evaluate the expected reward of an evaluation policy $\pi$ with the data collected by an behavior policy $\mu$. By defining $\mu_i$ to be $\mu(a_i|x_i)$, the histroy data can be represented as: $\{(x_i, a_i, r_i, \mu_i)\}_{i=1}^{\mathcal{N}_\mathcal{S}}$. The expected reward by executing policy $\pi$ takes the following form:

$$V(\pi; \mathcal{D}) := \mathbb{E}_{x \sim \mathcal{D}_\mathcal{X}(\cdot)} \mathbb{E}_{a \sim \pi(\cdot|x)} \mathbb{E}_{r \sim \mathcal{D}_\mathcal{R}(\cdot|x,a)}[r] \quad (1)$$

$$= \mathbb{E}_{x \sim \mathcal{D}_\mathcal{X}(\cdot)} \mathbb{E}_{a \sim \pi(\cdot|x)}[r^*(x, a)] \quad (2)$$

**Contextual Bandits under Distribution Shift**  We consider the problem of off-policy evaluation under domain/distribution shift. In particular, in addition to the difference in policies, the distribution intrinsic to the environment differs for the data collection phase and the evaluation phase. The problem is stated formally as follows:

In most off-policy evaluation settings, an oversimplified assumption is made – the underlying distribution remain unchanged, i.e., the distribution of context on which the behavior policy is executed (denoted as $\Pr_\mathcal{S}[x]$) is the same as the distribution of contexts on which the evaluation policy (denoted as $\Pr_\mathcal{T}[x]$) will be executed. We consider a more realistic assumption that the context distribution is shifted. We formally introduce the problem concretely as follows.

**Problem** [*Off-policy evaluation under distribution shift*] Given data sampled from the source distribution $\{(x_i, a_i, r_i, \mu_i)\}_{i=1}^{n_{\mathcal{S}}} \overset{\text{i.i.d.}}{\sim} (\mathcal{D}_{\mathcal{S}}, \mu)$ and contexts observed from the target domain $\{x_i'\}_{i=1}^{n_{\mathcal{T}}} \overset{\text{i.i.d.}}{\sim} \mathcal{D}_{\mathcal{T}}$, we aim to estimate

$$V(\pi; \mathcal{D}_{\mathcal{T}}) = \mathbb{E}_{(x,a,r) \sim (\mathcal{D}_{\mathcal{T}}, \pi)}[r] \quad (3)$$

An intuitive idea is to obtain an unbiased estimate of the reward by reweighting the samples from the source distribution using the inverse propensity score $\beta^\pi(x_i, a_i, r_i) := \frac{\Pr_{\mathcal{T}, \pi}[x_i, a_i, r_i]}{\Pr_{\mathcal{S}, \mu}[x_i, a_i, r_i]}$

$$\sum_{i=1}^{n_{\mathcal{S}}} \beta^\pi(x_i, a_i, r_i) r_i \quad (4)$$

However, the naive estimator in (4) usually exhibits high variance due to the high dimensionality of contexts $x$. We introduce a novel model called *Intent-shift model* (ISM) to address this issue.

### 2.2. Intent-shift Model

Our Intent-shift model considers a compact representation of the high dimensional context. With the hidden representation (called intent), we provide more robust estimators for off-policy evaluation under distribution shift.

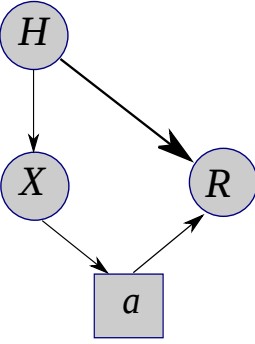

*Figure 1.* Graphical representation of the contextual bandit model with intents. The intent $h$ generates the observable features (contexts) $x$ used by the policy.

**Extended Contextual Bandits** We extend the traditional contextual bandit model with an auxillary variable $h$ which we call intent, since it governs the distribution over context and reward as user intent does in the application of recommendation or searching. This intent variable $h$ interacts with the context $x$, action $a$ and the reward $r$ via a directed (causal) graphical model in Figure 1. In this model, intent $h$ generates the observable features $x$ used by the policy, and it provides feedbacks $r$ to the learners which measures how appropriate the action $a$ taken by the policy is in addressing the user's intent.

It is not required that the intents are provided together with each triplets. But we assume there exists some off-the-shelf mappers that can provide high quality and conditionally independent predictions of the hidden intents, stated formally as follows:

**Definition 2.1** (Confusion Matrix). *Let $f_a, f_b, f_c$ be three mappers $f_a, f_b, f_c : \mathcal{X} \rightarrow \mathcal{H}$, each associated with a confusion matrix $C_a, C_b, C_c$ defined as $[C_g]_{ij} :=$*
$\Pr_{(x,h) \sim \mathcal{D}_{\mathcal{S}}}[f_g(x) = e_i | h = e_j] \overset{(a)}{=} \Pr_{(x,h) \sim \mathcal{D}_{\mathcal{T}}}[f_g(x) = e_i | h = e_j], \forall g \in \{a, b, c\}$.

Equality (a) above holds due to the intent shift assumption.

**Assumption 2.2** (Independence of mappers conditioned on the true label). *The decisions made by all classifiers are independent conditioned on the ground truth labels:* $\Pr[f_a(x) = e_i, f_b(x) = e_j, f_c(x) = e_k | h = e_l] = (C_a)_{il}(C_b)_{jl}(C_c)_{kl}$.

**Intent-shift Model** Here we propose the intent-shift model by analogy with the label-shift setting in domain adaptation for supervised learning. The hidden intents $h$ can be viewed as a compact low- dimensional representation of the high-dimensional observed contexts/features $x$. In particular, we consider categorical intent variable $h \in \mathcal{H} = \{1, 2, \cdots, K\}$. The assumption is stated formally as:

**Assumption 2.3** (Intent-shift). *The shift of the joint distribution can be ascribed to the shift of marginal distribution of the intent variable. That is, the conditional distribution $\Pr[x|h]$ and $\Pr[r|h]$ remains unchanged as the marginal distribution of $h$ shifts from source to target domain, which can be implied by the following decompostion of distribution:*

$$\Pr_{\mathcal{D}, \pi}[x, h, a, r] = \Pr_{\mathcal{D}}[h] \Pr[x|h] \Pr[r|h] \Pr_\pi[a|x] \quad (5)$$

*for a domain specific distribution $\mathcal{D} \in \{\mathcal{D}_{\mathcal{S}}, \mathcal{D}_{\mathcal{T}}\}$.*

The assumption indicates that the shift in the distribution of $x$ is due to the shift in the marginal distribution of intents ($\Pr_{\mathcal{S}}(h) \neq \Pr_{\mathcal{T}}(h)$) but how the intent generates user's search queries and other features remain unchanged ($\Pr_{\mathcal{S}}(x|h) = \Pr_{\mathcal{T}}(x|h)$). The assumption is similar to the label shift assumption in (Lipton et al., 2018)(Aziz-zadenesheli et al., 2018), where the conditional distribution of features given ground truth labels remains the same for the source and target distribution. These assumptions are reasonable, and in some sense without loss of generality due to Reinbach's Common Cause Principle ((Reichenbach, 1991)(Sober, 1988)), which says that "whenever there is a correlation between two variables ($x$ and $r$), there must be a common cause $h$."

## 3. Our Approach

Given a reasonable assumption on the distribution shift, now the question is how to come up with an estimator without density estimation over $x$. In other words, our goal is to use inverse propensity score that depends on the density estimation of $h$ to reweight the training examples. In this section, we present the IPS-based estimator for off-policy evaluation. Our estimator is based on a spectral method-based procedure for estimating the reweighting factors associate with the categorical intents. In this section, we will first provide the procedure for our estimator, then specified the estimator for reweighting factor and provide finite sample analysis.

### 3.1. IPS Estimator for Intent-shift Model

In this subsection, we describe the IPS estimator, assuming the availability of the estimation of reweighting factor. We further make the assumption that the supports of the distribution over $\mathcal{X}$ conditioned on different intents do not overlap:

**Assumption 3.1.** *For any $h, h' \in \mathcal{H}, h \neq h'$:*

$$supp(\Pr_{\mathcal{D}_\mathcal{X}}[\cdot|H = h]) \cap supp(\Pr_{\mathcal{D}_\mathcal{X}}[\cdot|H = h']) = \emptyset \quad (6)$$

The assumption indicates that, given the knowledge of the joint distribution, the corresponding $h$ can be uniquely assigned to an observed context with probability one.

Without loss of generality, let $h \in [K]$ be categorical variables.

To solve the problem of off-policy evaluation under distribution shift in Equation (3), we analyze the reweighting factor under the intent model with assumption 3.1:

**Lemma 3.2.** *For any $(x_i, a_i, r_i)$ sampled from the source distribution by behavior policy $\mu$:*

$$\beta^\pi(x_i, a_i, r_i) = \frac{\pi(a_i|x_i)}{\mu(a_i|x_i)} \frac{\Pr_\mathcal{T}[h_i]}{\Pr_\mathcal{S}[h_i]} \quad (7)$$

The proof for lemma 3.2 is included in Appendix A.

An intuitive plug-in estimator for off-policy evaluation (also see Algorithm 1) is:

$$\sum_{i=1}^{n_\mathcal{S}} \widehat{\beta}^\pi(x_i, a_i, r_i) r_i \quad \text{with } \widehat{\beta}^\pi(x_i, a_i, r_i) = \frac{\pi(a_i|x_i)}{\mu(a_i|x_i)} \frac{\widehat{\Pr}_\mathcal{T}[\widehat{h}_i]}{\widehat{\Pr}_\mathcal{S}[\widehat{h}_i]} \quad (8)$$

given triplets generated from the source domain $\{(x_i, a_i, r_i)\}_{i=1}^{n_\mathcal{S}}$ under behavior policy $\mu$. The associated intent is provided by querying any one of the off-the-shelf estimator. Notice that the estimator does not involve direct density estimation of context $x$.

Now the remaining problem is to estimate $\frac{\widehat{\Pr}_\mathcal{T}[\widehat{h}_i]}{\widehat{\Pr}_\mathcal{S}[\widehat{h}_i]}$ for any given $x_i$ from the source domain.

---

**Procedure 1** Proposed Spectral IPS Estimator

**Input:** $\{(x_i, a_i, r_i)\}_{i=1}^{n_\mathcal{S}} \sim \Pr_\mathcal{S}^\mu[x, a, r]$, $\{x'_i\}_{i=1}^{n_\mathcal{T}} \sim \Pr_\mathcal{T}[x]$ and mappers $f_a$, $f_b$ and $f_c$

**Output:** $\widehat{\beta}^\pi(x_i, a_i, r_i), \forall\{(x_i, a_i, r_i)\}_{i=1}^{n_\mathcal{S}}$

1: $\widehat{\psi} \leftarrow$ Procedure 3($\{x_i\}_{i=1}^{n_\mathcal{S}}, \{x'_i\}_{i=1}^{n_\mathcal{T}}, f_a, f_b, f_c$)
2: **for** $i = 1$ **to** $n_\mathcal{S}$ **do**
3: $\quad \widehat{h}_i \leftarrow f_c(x_i)$
4: $\quad \widehat{\beta}^\pi(x_i, a_i, r_i) \leftarrow \frac{\pi(a_i|x_i)}{\mu(a_i|x_i)} \widehat{\psi}(\widehat{h}_i)$
5: **end for**

---

### 3.2. Spectral-based Estimator for the reweighting factor

We now describe a spectral-based procedure of the reweighting factor (see also Algorithm 3). Let $\psi(h)$ denote the reweighting factor for intent $h$: $\frac{\Pr_\mathcal{T}[h]}{\Pr_\mathcal{T}[h]}$, for all $h$.

**Co-occurrence Matrices** Our method is based on direct estimation of the expected co-occurrence of predictions of the mappers, defined as follows:

**Definition 3.3.** *For all $\alpha, \beta \in \{a, b, c\}, \alpha \neq \beta$, the co-occurrence matrices and their empirical estimations is defined as:*

$$(M_\mathcal{S}^{\alpha\beta})_{ij} := \mathbb{E}_{x \sim \mathcal{D}_\mathcal{S}}[\mathbb{1}[f_\alpha(x) = e_i, f_\beta(x) = e_j]] \quad (9)$$

$$(M_\mathcal{T}^{\alpha\beta})_{ij} := \mathbb{E}_{x \sim \mathcal{D}_\mathcal{T}}[\mathbb{1}[f_\alpha(x) = e_i, f_\beta(x) = e_j]] \quad (10)$$

$$(\widehat{M}_\mathcal{S}^{\alpha\beta})_{ij} := \frac{1}{n_\mathcal{S}} \sum_{k=1}^{n_\mathcal{S}} \mathbb{1}[f_\alpha(x_k) = e_i \wedge f_\beta(x_k) = e_j]$$

$$(11)$$

$$(\widehat{M}_\mathcal{T}^{\alpha\beta})_{ij} := \frac{1}{n_\mathcal{T}} \sum_{k=1}^{n_\mathcal{T}} \mathbb{1}[f_\alpha(x_k) = e_i \wedge f_\beta(x_k) = e_j]$$

$$(12)$$

An important observation is that the second order co-occurrence matrix of predictions are simultaneously diagonalizable, under the following conditional independence property of the mappers.

**Lemma 3.4** (Simultaneous Diagonalization)**.** *Under the assumption 2.2, the second order co-occurence matrix of predictions could be factorized as the confusion matrices and the marginal distribution of $h$:*

$$M_\mathcal{S}^{ab} = C_a \Lambda_\mathcal{S} C_b^\top \quad M_\mathcal{S}^{bc} = C_b \Lambda_\mathcal{S} C_c^\top \quad M_\mathcal{S}^{ac} = C_a \Lambda_\mathcal{S} C_c^\top$$

$$M_\mathcal{T}^{ab} = C_a \Lambda_\mathcal{T} C_b^\top \quad M_\mathcal{T}^{bc} = C_b \Lambda_\mathcal{T} C_c^\top \quad M_\mathcal{T}^{ac} = C_a \Lambda_\mathcal{T} C_c^\top$$

*where $\Lambda_\mathcal{S} := [\Pr_\mathcal{S}[h = 1], \ldots, \Pr_\mathcal{S}[h = K]]$ and $\Lambda_\mathcal{T} := [\Pr_\mathcal{T}[h = 1], \ldots, \Pr_\mathcal{T}[h = K]]$.*

Such observation allows us to learn the shift in the distribution over $h$ through a joint-diagonalization procedure on a symmetrized second moment.

**Lemma 3.5** (Symmetrization). *If we define $M_{\mathcal{S}} := M_{\mathcal{S}}^{cb}(M_{\mathcal{S}}^{ab})^{\dagger}M_{\mathcal{S}}^{ac}$ on the source domain, the obtained second order co-occurrence matrix of predictions $M_{\mathcal{S}}$ is symmetric and diagonalizable via $C_c$: $M_{\mathcal{S}} = C_c\Lambda_{\mathcal{S}}C_c^{\top}$. Similar statement holds for $M_{\mathcal{T}}$ on the target domain.*

**Direct estimation for $\psi(h)$ through joint diagonalization** Given the estimation for the symmetric matrices $M_{\mathcal{S}}$ and $M_{\mathcal{T}}$, a whitening step yield a matrix with the reweighting factors of intents as its eigenvalues:

**Lemma 3.6** (Whitening). *Let $W_{\mathcal{S}} = U_{\mathcal{S}}\Sigma_{\mathcal{S}}^{-\frac{1}{2}}$ be the whitening matrix such that $W_{\mathcal{S}}^{\top}M_{\mathcal{S}}W_{\mathcal{S}} = I$, then $W_{\mathcal{S}}^{\top}M_{\mathcal{T}}W_{\mathcal{S}}$ admits an eigen-decomposition with $\{\psi(h)\}_{i=1}^{K}$ as the eigenvalues.*

This property allows identification of the ratio of the marginal distribution of $h$ through eigen-decomposition of the whitened matrix $\widehat{W_{\mathcal{S}}}^{\top}\widehat{M_{\mathcal{T}}}\widehat{W_{\mathcal{S}}}$. The proof for Lemma 3.4, 3.5, 3.6 is in Appendix A.

**Alignment Issue** The joint diagonalization procedure provides an estimation of the shift in distribution of intent $\{\psi(h) = \frac{p_{\mathcal{T}}(h)}{p_{\mathcal{S}}(h)}|h \in \mathcal{H}\}$, which is an unordered set of reweighting factors. The recovery procedure suffers from mis-alignment.

We align the eigenvalues with the intents through an estimation of the confusion matrices, by permuting the order of the elements of $\widehat{\psi}$ with a permutation matrix $T^*$ that permutes the confusion matrix $C_c$ to be diagonal-dominant:

$$T^* \leftarrow \arg\min_{T}\|I - C_cT\|. \tag{13}$$

The estimation of the confusion matrix follows from procedure 2, where $\widehat{\psi}(h)$ corresponds to the $h^{\text{th}}$ column of $C_c$. Notice that the estimation error of the reweighting factor does not depend on the estimation error of the confusion matrices, thus does not contribute to the sample complexity.

---

**Procedure 2** Confusion Matrix Estimation

**Input:** Whitened Symmetrized second order statistics $\widehat{M} = \widehat{W}_{\mathcal{S}}^{\top}\widehat{M}_{\mathcal{T}}\widehat{W}_{\mathcal{S}}$

**Output:** Estimation of confusion matrix $\widehat{C}_c$
  1: $\tilde{c}_j \leftarrow j^{th}$ Eigenvector$(\widehat{W}_{\mathcal{S}}^{\top}\widehat{M}_{\mathcal{T}}\widehat{W}_{\mathcal{S}})$
  2: $\widehat{C}_c \leftarrow [\tilde{c}_j/\|\tilde{c}_j\|_1]_{j=1}^{K}$

---

**Procedure 3** Estimator for $\psi$ via Joint Diagonalization

**Input:** $\{x_i\}_{i=1}^{n_{\mathcal{S}}} \sim \Pr_{\mathcal{S}}[x]$, $\{x_i'\}_{i=1}^{n_{\mathcal{T}}} \sim \Pr_{\mathcal{T}}[x]$ and mappers $f_a$, $f_b$ and $f_c$

**Output:** $\widehat{\psi} = [\hat{\psi}(1), \dots, \hat{\psi}(h), \hat{\psi}(K)]^{\top}$
  1: Compute $\widehat{M}_{\mathcal{S}}^{ab}$, $\widehat{M}_{\mathcal{S}}^{bc}$ and $\widehat{M}_{\mathcal{S}}^{ac}$
  2: Compute $\widehat{M}_{\mathcal{T}}^{ab}$, $\widehat{M}_{\mathcal{T}}^{bc}$ and $\widehat{M}_{\mathcal{T}}^{ac}$
  3: Compute $M_{\mathcal{S}}$ and $M_{\mathcal{T}}$

$$\widehat{M}_{\mathcal{S}} = \widehat{M}_{\mathcal{S}}^{cb}(\widehat{M}_{\mathcal{S}}^{ab})^{\dagger}\widehat{M}_{\mathcal{S}}^{ac} \tag{14}$$
$$\widehat{M}_{\mathcal{T}} = \widehat{M}_{\mathcal{T}}^{cb}(\widehat{M}_{\mathcal{T}}^{ab})^{\dagger}\widehat{M}_{\mathcal{T}}^{ac} \tag{15}$$

  4: $U_{\mathcal{S}} \leftarrow$ Eigenvectors$(\widehat{M}_{\mathcal{S}})$, $\Sigma_{\mathcal{S}} \leftarrow$ Eigenvalues$(\widehat{M}_{\mathcal{S}})$
  5: $\widehat{W}_{\mathcal{S}} \leftarrow U_{\mathcal{S}}\Sigma_{\mathcal{S}}^{-\frac{1}{2}}$
  6: $\hat{\psi} \leftarrow$ Eigenvalues$(\widehat{W}_{\mathcal{S}}^{\top}\widehat{M}_{\mathcal{T}}\widehat{W}_{\mathcal{S}})$,
  7: $\widehat{C}_c \leftarrow$ Procedure $2(\widehat{W}_{\mathcal{S}}^{\top}\widehat{M}_{\mathcal{T}}\widehat{W}_{\mathcal{S}})$
  8: $T^* \leftarrow \arg\min_{T}\|I - C_cT\|$
  9: $\hat{\psi} \leftarrow T^*\hat{\psi}$

---

### 3.3. Finite Sample Analysis of $\|\hat{\psi} - \psi\|_{\infty}$

Now we present our main result on the finite sample analysis of $\|\hat{\psi} - \psi\|_{\infty}$m for $\hat{\psi}$ given by Procedure 3:

**Theorem 3.7** (Sample Complexity). *There exists $N_0, C \in \mathbb{Z}_+$ such that for all sample size $\mathcal{N}_{\mathcal{S}}, \mathcal{N}_{\mathcal{T}} \geq \max\{N_0, \frac{C}{\epsilon^2}\log\frac{6}{\delta}\}$, with probability $\geq 1 - \delta$, our algorithm yields $\hat{\psi}$ such that $\|\psi - \hat{\psi}\|_{\infty} \leq \epsilon$.*

Here, the constant $C$ and $N_0$ depends on the smallest singular value of the confusion matrices. The proof of the main theorem is in appendix C.

**Remark.** *Our spectral estimator $\hat{\psi}$ for the reweighting factor $\psi$ is unbiased and convergences to the true reweighting factor $\psi$ asymptotically. With high probability greater than $1 - \delta$, the difference between the estimated and true reweighting factor is bounded by a small prevision $\epsilon$ as long as there are more than $\max\{N_0, \frac{C}{\epsilon^2}\log\frac{6}{\delta}\}$ number of observations.*

## 4. Analyses and Gaurantees

In this section we provide bias and mean square error of the Spectral-based IPS Estimatorin Procedure 1, followed by our main result, the finite sample analysis for the estimation of the reweighting factor. For the simplicity of notation, $p_{\mathcal{S}}$ will be used to denote $\Pr_{\mathcal{S}}[h]$ and $p_{\mathcal{T}}$ will be used to denote $\Pr_{\mathcal{T}}[h]$.

### 4.1. Bias and MSE Analysis

Since perfect mapping from the context to the hidden factor is not always available, the bias is unavoidable. We first provide a bound on the absolute value of the bias of the

estimator.

**Theorem 4.1** (Bias of the Estimator). *The absolute value of the bias of the estimator is bounded by:*

$$\left| \mathbb{E}_{x\sim\mathcal{D}_\mathcal{S}, a\sim\mu(\cdot|x)}[(\beta(x,a,r) - \hat{\beta}(x,a,r))r] \right| \leq$$

$$\left( \frac{2\|p_\mathcal{S} - p_\mathcal{T}\|_2}{p_{\min}^2} \mathbb{E}_{x\sim\mathcal{D}_\mathcal{S}}\left[ \mathbf{1}[h(x) \neq \hat{h}(x)] \right] + \|\psi - \hat\psi\|_\infty \right) R_{\max}$$

**Remark.** *The second term converges to zero with high probability as the number of samples increases, while the first term is a systematic error depending on the quality of the mappers $f_a$, $f_b$ and $f_c$.*

Our proposed estimator balances between the bias and variance. Now we provide an upper bound on the mean squared error of the estimator which considers the bias and variance jointly. Let $\mathbf{Var}_{\mathsf{IPS}}$ denote the variance induced by *importance sampling* between target and behavior policy under the same source domain (i.e., no distributional shift). Therefore this variance term $\mathbf{Var}_{\mathsf{IPS}} = |\mathbb{E}_{x\sim\mathcal{D}_\mathcal{S}, a\sim\mu(\cdot|x)}[(\mathbb{E}_{x'\sim\mathcal{D}_\mathcal{S}, a'\sim\mu(\cdot|x')}[\beta(x',a')r(x',a')] - \beta(x,a)r(x,a))^2]|$ is unavoidable.

**Theorem 4.2** (Mean Square Error of the Estimator). *The mean squared error of the estimator is upper bounded by*

$$MSE \leq 2\Big( \mathbf{Var}_{\mathsf{IPS}} + \Big[ b\mathbb{E}_{x\sim\mathcal{D}_\mathcal{S}}\left[ \mathbf{1}[h(x) \neq \hat{h}(x)] \right]$$
$$+ \|\psi - \hat\psi\|_\infty^2 \Big] D_{\chi^2}(\pi\|\mu) R_{\max}^2 \Big) \quad (16)$$

*where* $b = \frac{\|p_\mathcal{T} - p_\mathcal{S}\|^2}{p_{\min}^4} + \frac{2\|p_\mathcal{T} - p_\mathcal{S}\|}{p_{\min}^2}\|\psi - \hat\psi\|_\infty$, $p_{\min} := \inf_{h\in\mathcal{H}} p_\mathcal{S}(h)$*, and the $\chi^2$-divergence is defined as:*

$$D_{\chi^2}(\pi\|\mu) := \mathbb{E}_\mu\left[ \left(\frac{\pi}{\mu}\right)^2 \right] \quad (17)$$

The proof is in Appendix B.

**Remark.** *As shown in Theorem 4.2 (1) The MSE of our estimator inevitably consists of the Importance Sampling variance with no distributional shift, which is expected as the considered problem is strictly more challenging. (2) The MSE of our estimator is smaller if the quality of the mappers $f_a$, $f_b$ and $f_c$ is better, as $\mathbb{E}_{x\sim\mathcal{D}_\mathcal{S}}[\mathbf{1}[h(x) \neq \hat{h}(x)]]$ can be upper-bounded by $1 - c_{\min}$, where $c_{\min}$ is the smallest element on the diagonal of the confusion matrix. (3) The MSE of our estimator depends on the quality of our estimation of reweighting factor $\psi$, which can be arbitrarily small with enough observations. (4) A smaller $\chi^2$-divergence between behavior and target policy results in a better MSE.*

## 5. Experiments

We design experiments in this section to evaluate the performance of our proposed estimator on MNIST dataset under distributional shift.

### 5.1. Experimental Setup

The data is created by sampling from the MNIST dataset according to certain distributions, with the following conversion to bandit setting:

**Context, Action, Reward and Policy.** The context are the images of digits in MNIST dataset, and the actions space is the label prediction of the images. A policy provides a probabilistic prediction of the label of the image, and the reward is a function of the indicator whether the prediction is correct.

**Behavior Policy $\mu$ and Target Policy $\pi$.** We considered the random prediction policy $\mu(a|x) = 0.1$, $\forall a, x$ as the behavior policy to simulate the exploration phase in online-learning setting. For the target policy under which we need to evaluate the reward, we consider an adapted policy from classifiers for MNIST dataset. To be specific, we use the outcome of the softmax layer as $\pi(a|x)$ given the input of features.

**Cost-sensitive Reward Model.** A cost-sensitive reward model is considered for generality (allowing the general cost-sensitive classification). The reward model is $r(x, a, h) = c(\cdot)^\top a(\cdot) = c_h \mathbb{1}[a = h]$, where $c(\cdot) = [c_1, c_2, \ldots, c_{10}]$. In our experiment, we consider a special case by setting $c(\cdot)$ to be $[1, 2, 3, 4, 5, 6, 7, 8, 9, 10]$ without loss of generality.

**Baselines.** We compare our results with two baselines, the first one one is standard inverse propensity score estimator without taking the distribution shift into consideration(Dudík et al., 2014), where we average the reward only reweighted by the IPS for policies, the population version of which is the expected reward under source distribution. By comparison with such baseline, we show how distribution shift makes the problem of off-policy evaluation more difficult. The second baseline is kernel mean matching that does not imposing intent shift assumption(Gretton et al., 2009). For tractability, principal component analysis is used for dimension reduction (from $784 - d$ to $30 - d$).

### 5.2. Results on Varying Target Intent Distributions

To compare the performance of our estimator with baselines over a wide range of different target distributions, we consider a general setting where the target distributions are drawn from the a dirichlet distribution controlled by a parameter $\alpha$. For small $\alpha$s, the drawn target distribution are close to a singleton and for large $\alpha$s the target distribution is close to a uniform distribution over $\mathcal{H}$. We will run experiments on different $\alpha$s.

In Figure 2a we compare our method with baseline and kernel mean matching (KMMD) method(Gretton et al., 2009) and demonstrate that our estimator outperforms the base-

line models for a range of different target distributions controlled by different $\alpha$s. As illustrated in the plot, due to the high-dimensionality of the context space, it is hard for KMMD to achieve a good estimation of the expected reward without further assumption on the distribution shift.

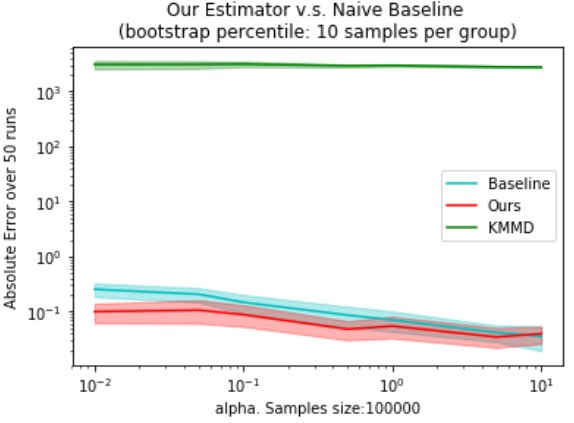

**(a)** Comparison with baselines

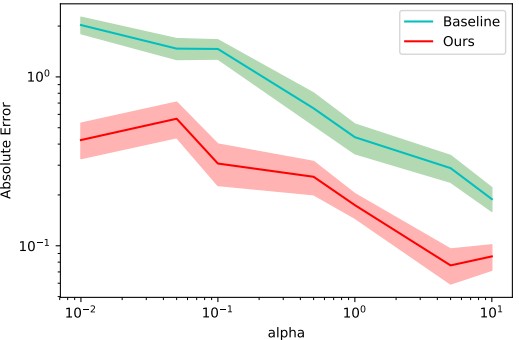

**(b)** Zoomed-in comparison with unweighted average baseline

*Figure 2.* Randomly Drawn Distributions: In this figure we compare the kernel mean matching method, the standard IPS baseline without correcting the distribution shift, and our method.

In Figure 3a we plot the estimation error of our estimator for the distribution shift. As shown in Figure 3a, our estimator performs better for larger larger, i.e., our estimator performs better when the target distribution is more spread out.

### 5.3. A Hard Case Analysis: Categorical Target Intent Distribution

Now we analyze the scenario of categorical target intent distribution, under which our estimators performs worst (small $\alpha$s), and we do a detailed comparison against the best baseline — unweighted average.

We first illustrate how the distribution shift on context space affects the off-policy evaluation of cumulative re-

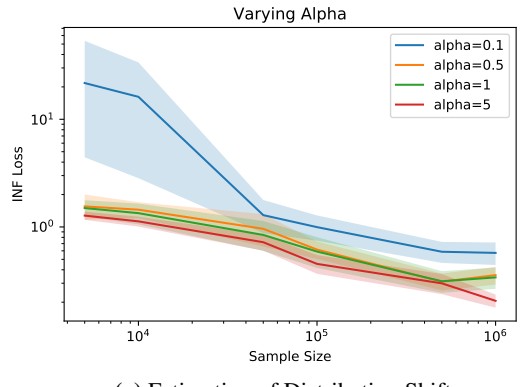

**(a)** Estimation of Distribution Shift

*Figure 3.* The performance of our estimator under different target intent distributions.

ward. We fix the source intent distribution and consider a fixed set of target intent distribution, the support of which is a fixed $\{h\} \subset \mathcal{H}$. Specifically, the source distribution on intent $h$ is uniform over $\mathcal{H}$: $\Pr_{\mathcal{S}}[h] = [0.1, \ldots, 0.1]^\top$ and the target distribution on intent $h$ concentrates on one category: $\Pr_{\mathcal{T}}[h] = [0, \ldots, 1, 0, \ldots, 0]^\top$.

As shown in Figure 4, our estimator provides a good evaluation of the reward model under distribution shift. The $x$-axis is the support of target intent distribution. In Figure 4(a) we compare against the first baseline, denoted as 'muS', the ground truth 'piT', and the estimator we propose 'Our Estimates'.

From Figure 4(b) we see that under this hard scenario, our estimator outperforms the baseline when the ground truth category are the ones that result in a large difference between the reward in source and the reward in target domain, as expected.

## 6. Conclusion

We studied the hardness of correcting distribution shift in the contextual bandit setting. We provided an estimation procedure with finite sample analysis and MSE analysis. Our estimator outperforms the kernel mean matching baseline without assumptions on the distribution shift and with a standard inverse propensity score without considering the distribution shift.

Our estimator can be further improved as the current version requires knowledge of the intent space (i.e. the intent space is not truly latent). One future direction is extending to a more general setting that does not involve this knowledge and learns the latent space based on data.

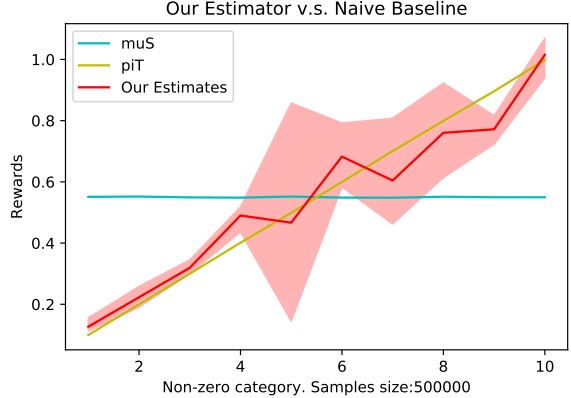

**(a)** Estimation of expected rewards

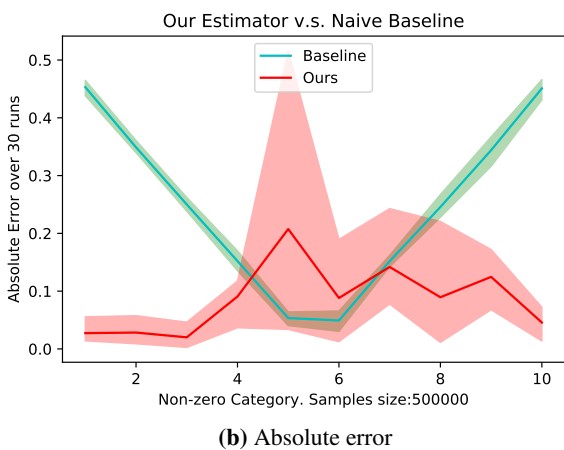

**(b)** Absolute error

*Figure 4.* Performance comparison for categorical target distribution. **(a)** average absolute error of reward estimation and **(b)** estimation of expected rewards over 30 experiments for each target distribution.

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

# Appendix: A Spectral Method for Off-Policy Evaluation in Contextual Bandits under Distribution Shift

## A. Spectral Method for Estimating Reweighting Factors

### A.1. Proof for Lemma 3.2

We define a $q(x_i)$ as a distribution vector of length $K$, where the $h^{\text{th}}$ element of vector $q(x_i)$ is $[q(x_i)]_h = \frac{\Pr[x_i|h]}{\sum_h \Pr[x_i|h]}$.

$$\beta^\pi(x_i, a_i, r_i) = \quad \frac{\Pr_{\mathcal{T},\pi}[x_i, a_i, r_i]}{\Pr_{\mathcal{S},\mu}[x_i, a_i, r_i]} \tag{18}$$

$$\overset{\text{Intent-shift}}{=} \frac{\pi(a_i|x_i)}{\mu(a_i|x_i)} \frac{\sum_h \Pr_{\mathcal{T}}[h]\Pr[x_i|h]}{\sum_h \Pr_{\mathcal{S}}[h]\Pr[x_i|h]} \tag{19}$$

$$= \frac{\pi(a_i|x_i)}{\mu(a_i|x_i)} \frac{\mathbb{E}_{q(x)}[\Pr_{\mathcal{T}}[h]]}{\mathbb{E}_{q[x]}[\Pr_{\mathcal{S}}[h]]} \tag{20}$$

$$\overset{\text{pre-imaging}}{=} \frac{\pi(a_i|x_i)}{\mu(a_i|x_i)} \frac{\Pr_{\mathcal{T}}[h_i]}{\Pr_{\mathcal{S}}[h_i]} \tag{21}$$

Equation (21) holds since the distribution vector $q(x_i)$ is a one-hot encoding vector, and $\widehat{h}_i^{\text{th}}$ element is 1.

### A.2. Proof for Lemma 3.4

*Proof.* Due to the conditional independence property of the mappers, the probability of joint prediction conditioned on the ground truth label can be decomposed as a product of distributions.

$$(M_{\mathcal{S}}^{(ab)})_{ij} := \mathbb{E}_{(x,y)\sim\mathcal{S}}[\mathbf{1}[f_a(x) = e_i \wedge f_b(x) = e_j]] \tag{22}$$

$$= \sum_{l=1}^K \Pr_{\mathcal{S}}[h = l]\Pr_{\mathcal{S}}[f_a(x) = e_i \wedge f_b(x) = e_j|h = l] \tag{23}$$

$$= \sum_{l=1}^K \Pr_{\mathcal{S}}[h = l](C_a)_{il}(C_b)_{jl} \tag{24}$$

$$= (C_a\Lambda_{\mathcal{S}}C_b^\top)_{ij} \tag{25}$$

The proof for the target domain is identical. $\qquad\square$

### A.3. Proof for Lemma 3.5

*Proof.* In order to symmetrize the second order statistics, we need to find matrix $W_a$ and $W_b$ such that:

$$W_aC_a = C_c; \quad W_bC_b = C_c \tag{26}$$

Equation 26 leads to:

$$W_aM_{\mathcal{T}}^{ab} = W_aC_a\Lambda_{\mathcal{T}}C_b^\top = C_c\Lambda_{\mathcal{T}}C_b^\top = M_{\mathcal{T}}^{cb} \tag{27}$$

$$W_bM_{\mathcal{T}}^{ba} = W_bC_b\Lambda_{\mathcal{T}}C_a^\top = C_c\Lambda_{\mathcal{T}}C_a^\top = M_{\mathcal{T}}^{ca} \tag{28}$$

$$\tag{29}$$

By defining the symmetrization matrix as following

$$W_a = M_{\mathcal{T}}^{cb}(M_{\mathcal{T}}^{ab})^{-1} \tag{30}$$

$$W_b = M_{\mathcal{T}}^{ca}(M_{\mathcal{T}}^{ba})^{-1} \tag{31}$$

The result of the lemma follows directly. $\qquad\square$

### A.4. Proof for Lemma 3.6

*Proof.* In the whitening step, we first find a whitening matrix $\hat{W}$ for the second order moment on source domain such that:

$$W^\top M_{\mathcal{S}} W = I \tag{32}$$

Assume matrix $M_{\mathcal{S}}$ allows an eigen-decomposition:

$$M_{\mathcal{S}} := C_c \Lambda_{\mathcal{S}} C_c^\top = U_{\mathcal{S}} \Sigma_{\mathcal{S}} U_{\mathcal{S}}^\top \tag{33}$$

then the whitening matrix $W$ takes the form:

$$W = U_{\mathcal{S}} \Sigma_{\mathcal{S}}^{-\frac{1}{2}} \tag{34}$$

Given the following two symmetrized statistcs from the source and target domain:

$$M_{\mathcal{S}} = C \Lambda_{\mathcal{S}} C^\top = \sum_{r=1}^{R} \lambda_r^{(\mathcal{S})} \mu_r^{\otimes 2} \tag{35}$$

$$M_{\mathcal{T}} = C \Lambda_{\mathcal{T}} C^\top = \sum_{r=1}^{R} \lambda_r^{(\mathcal{T})} \mu_r^{\otimes 2} \tag{36}$$

where $\mu_r$ is the $r$-th column of the matrix $C$ (and the conditional probability of ground truth class $r$), we have:

$$W^\top M_{\mathcal{T}} W = \sum_{r=1}^{R} \frac{\lambda_r^{(\mathcal{T})}}{\lambda_r^{(\mathcal{S})}} \tilde{\mu}_r^{\otimes 2} = \sum_{r=1}^{R} \psi_r \tilde{\mu}_r^{\otimes 2} \tag{37}$$

where $\tilde{\mu}_r^\top \tilde{\mu}_\rho = \delta_{r\rho}$ and $\mu_r = \frac{(\hat{W}^{-1})\tilde{\mu}_r}{|(\hat{W}^{-1})\tilde{\mu}_r|}$, $\forall r, \rho \in \{1, \cdots, K\}$. And $\{\psi_r\}_{r=1}^{R}$ is a permutation of the reweighting factors. $\square$

## B. Bias and MSE analysis

### B.1. Estimating the shift in context

In this subsection we provide the proof for the estimation error given the perfect distribution shift on the hidden factor space.

**Lemma B.1.**

$$\left| \frac{\mathbb{E}_{h\sim\hat{q}(\cdot|x)}[p_{\mathcal{T}}(h)]}{\mathbb{E}_{h\sim\hat{q}(\cdot|x)}[p_{\mathcal{S}}(h)]} - \frac{\mathbb{E}_{h\sim q(\cdot|x)}[p_{\mathcal{T}}(h)]}{\mathbb{E}_{h\sim q(\cdot|x)}[p_{\mathcal{S}}(h)]} \right| \leq \frac{\|q - \hat{q}\|_2 \|p_{\mathcal{S}} - p_{\mathcal{T}}\|_2}{\mathbb{E}_q[p_{\mathcal{S}}]\mathbb{E}_{\hat{q}}[p_{\mathcal{S}}]} \leq \frac{\|q - \hat{q}\|_2 \|p_{\mathcal{S}} - p_{\mathcal{T}}\|_2}{p_{\min}^2} \tag{38}$$

*where $p_{\min}$ is defined as $\min_{h\in\mathcal{H}} p_{\mathcal{S}}(h)$.*

*Proof.* We use the following short hands for briefness:

$$\left| \frac{\mathbb{E}_{h\sim\hat{q}(\cdot|x)}[p_{\mathcal{T}}(h)]}{\mathbb{E}_{h\sim\hat{q}(\cdot|x)}[p_{\mathcal{S}}(h)]} - \frac{\mathbb{E}_{h\sim q(\cdot|x)}[p_{\mathcal{T}}(h)]}{\mathbb{E}_{h\sim q(\cdot|x)}[p_{\mathcal{S}}(h)]} \right| := \left| \frac{\mathbb{E}_{\hat{q}}[p_{\mathcal{T}}]}{\mathbb{E}_{\hat{q}}[p_{\mathcal{S}}]} - \frac{\mathbb{E}_q[p_{\mathcal{T}}]}{\mathbb{E}_q[p_{\mathcal{S}}]} \right| \tag{39}$$

$$\left| \frac{\mathbb{E}_{\hat{q}}[p_{\mathcal{T}}]}{\mathbb{E}_{\hat{q}}[p_{\mathcal{S}}]} - \frac{\mathbb{E}_q[p_{\mathcal{T}}]}{\mathbb{E}_q[p_{\mathcal{S}}]} \right| \leq \frac{1}{\mathbb{E}_q[p_{\mathcal{S}}]\mathbb{E}_{\hat{q}}[p_{\mathcal{S}}]} |\mathbb{E}_q[p_{\mathcal{T}}]\mathbb{E}_{\hat{q}}[p_{\mathcal{S}}] - \mathbb{E}_q[p_{\mathcal{S}}]\mathbb{E}_{\hat{q}}[p_{\mathcal{T}}]| \tag{40}$$

Using the notation $\delta p := p_{\mathcal{T}} - p_{\mathcal{S}}$ and $\delta q := \hat{q} - q$, The second term can be rewritten as:

$$|\mathbb{E}_q[p_{\mathcal{T}}]\mathbb{E}_{\hat{q}}[p_{\mathcal{S}}] - \mathbb{E}_q[p_{\mathcal{S}}]\mathbb{E}_{\hat{q}}[p_{\mathcal{T}}]| \tag{41}$$

$$= \langle q, p_{\mathcal{S}} \rangle \left| \langle \delta q \left| I - \frac{1}{\langle q, p_{\mathcal{S}} \rangle} |q\rangle\langle p_{\mathcal{S}}| \right| \delta p \rangle \right| \tag{42}$$

$$\leq \|\delta q\|_2 \|\delta p\|_2 \text{(By Lemma B.2)} \tag{43}$$

$$\leq 2\sqrt{D_{KL}(\hat{q}\|q)D_{KL}(p_{\mathcal{T}}\|p_{\mathcal{S}})} \text{ (By Pinkster Inequality)} \tag{44}$$

$\square$

**Lemma B.2.** *The singular values of $\boldsymbol{R} := \boldsymbol{I}\langle \boldsymbol{u}, \boldsymbol{v}\rangle - \boldsymbol{u}\boldsymbol{v}^\top$ are 1 and $-\langle \boldsymbol{u}, \boldsymbol{v}\rangle-$, where $\boldsymbol{u}$ and $\boldsymbol{v}$ are unit vectors.*

*Proof.* First we notice that for any unit vector $\boldsymbol{w}$ orthogonal to both $\boldsymbol{u}$ and $\boldsymbol{v}$, $\boldsymbol{R}\boldsymbol{w} = \boldsymbol{R}^\top \boldsymbol{w} = \langle \boldsymbol{u}, \boldsymbol{v}\rangle \boldsymbol{w}$. From then on we can just focus vectors in the space spanned by $\boldsymbol{u}$ and $\boldsymbol{v}$. Consider the symmetric matrix $\boldsymbol{R}^\top \boldsymbol{R}$ and a unit vector $\boldsymbol{w} = \alpha_1 \boldsymbol{u} + \alpha_2 \boldsymbol{v}$.

$$\boldsymbol{R}^\top \boldsymbol{R}\boldsymbol{w} = \alpha_2 (\boldsymbol{I}\langle \boldsymbol{u}, \boldsymbol{v}\rangle - \boldsymbol{v}\boldsymbol{u}^\top)(-\boldsymbol{u} + \langle \boldsymbol{u}, \boldsymbol{v}\rangle \boldsymbol{v}) \tag{45}$$

$$= \alpha_2 (-\langle \boldsymbol{u}, \boldsymbol{v}\rangle \boldsymbol{u} + \boldsymbol{v}) \tag{46}$$

We have that $\alpha_1 = -\langle \boldsymbol{u}, \boldsymbol{v}\rangle \alpha_2$ if $\boldsymbol{w}$ is an eigenvector of $\boldsymbol{R}^\top \boldsymbol{R}$, and the eigenvalue is 1. $\square$

### B.2. Bias of the estimator

We first exam the bias of the estimator:

$$|\boldsymbol{bias}| = |\mathbb{E}_{x \sim \mathcal{D}_\mathcal{S}, a \sim \mu(\cdot|x)}[(\beta(x, a) - \hat{\beta}(x, a))r(x, a)]| \tag{47}$$

$$= |\mathbb{E}_{x \sim \mathcal{D}_\mathcal{S}, a \sim \mu(\cdot|x)}[(\beta(x, a) - \hat{\beta}(x, a))]|R_{max} \tag{48}$$

$$= |\mathbb{E}_{x \sim \mathcal{D}_\mathcal{S}, a \sim \mu(\cdot|x)}[\frac{\pi(a|x)}{\mu(a|x)}(\gamma(x) - \hat{\gamma}(x))]|R_{max} \tag{49}$$

$$= (|\mathbb{E}_{x \sim \mathcal{D}_\mathcal{S}, a \sim \pi(\cdot|x)}[\gamma(x) - \hat{\gamma}(x)]|)R_{max} \tag{50}$$

$$\leq (|\mathbb{E}_{x \sim \mathcal{D}_\mathcal{S}, a \sim \pi(\cdot|x)}[\gamma(x) - \hat{\gamma}(x, \psi)]| \tag{51}$$

$$+ |\mathbb{E}_{x \sim \mathcal{D}_\mathcal{S}, a \sim \pi(\cdot|x)}[\hat{\gamma}(x, \hat{\psi}) - \hat{\gamma}(x, \psi)]|)R_{max} \tag{52}$$

$$\leq (|\mathbb{E}_{x \sim \mathcal{D}_\mathcal{S}, a \sim \pi(\cdot|x)}[\gamma(x) - \hat{\gamma}(x, \psi)]| \tag{53}$$

$$+ |\mathbb{E}_{x \sim \mathcal{D}_\mathcal{S}, a \sim \pi(\cdot|x)}[\hat{\gamma}(x, \hat{\psi}) - \hat{\gamma}(x, \psi)]|)R_{max} \tag{54}$$

$$\leq (\frac{2\|p_\mathcal{S} - p_\mathcal{T}\|_2}{p_{\min}^2}\mathbb{E}_{x \sim \mathcal{D}_\mathcal{S}}[\mathbf{1}[h(x) \neq \hat{h}(x)]] + \|\psi - \hat{\psi}\|_\infty)R_{max} \tag{55}$$

The second term converges to zero with high probability as the number of samples increases, while the first term is a systematic error depending on the quality of the mapper/classifier.

### B.3. MSE Analysis

$$|\boldsymbol{MSE}| = |\mathbb{E}_{x \sim \mathcal{D}_\mathcal{S}, a \sim \mu(\cdot|x)}[(\mathbb{E}_{x' \sim \mathcal{D}_\mathcal{S}, a' \sim \mu(\cdot|x')}[\beta(x', a')r(x', a')] - \hat{\beta}(x, a)r(x, a))^2]| \tag{56}$$

$$= |\mathbb{E}_{x \sim \mathcal{D}_\mathcal{S}, a \sim \mu(\cdot|x)}[(\mathbb{E}_{x' \sim \mathcal{D}_\mathcal{S}, a' \sim \mu(\cdot|x')}[\beta(x', a')r(x', a')] - \beta(x, a)r(x, a) \tag{57}$$

$$+ \beta(x, a)r(x, a) - \hat{\beta}(x, a)r(x, a))^2]| \tag{58}$$

$$\leq 2(\textbf{Var}_{\mathsf{IPS}} + \mathbb{E}_{x \sim \mathcal{D}_\mathcal{S}, a \sim \mu(\cdot|x)}[(\beta(x, a)r(x, a) - \hat{\beta}(x, a)r(x, a))^2]) \tag{59}$$

The first term is the variance due to Importance Sampling which cannot be avoided:

$$\textbf{Var}_{\mathsf{IPS}} := |\mathbb{E}_{x \sim \mathcal{D}_\mathcal{S}, a \sim \mu(\cdot|x)}[(\mathbb{E}_{x' \sim \mathcal{D}_\mathcal{S}, a' \sim \mu(\cdot|x')}[\beta(x', a')r(x', a')] - \beta(x, a)r(x, a))^2]| \tag{60}$$

The second term can be bounded as:

$$\mathbb{E}_{x \sim \mathcal{D}_\mathcal{S}, a \sim \mu(\cdot|x)}[(\beta(x, a)r(x, a) - \hat{\beta}(x, a)r(x, a))^2] \tag{61}$$

$$= \mathbb{E}_{x \sim \mathcal{D}_\mathcal{S}}[(\gamma(x) - \hat{\gamma}(x, \hat{\psi}))^2 \mathbb{E}_{a \sim \pi(\cdot|x)}[\frac{\pi(a|x)}{\mu(a|x)}r(x, a)^2]] \tag{62}$$

$$\leq \left[(\frac{\|p_\mathcal{T} - p_\mathcal{S}\|^2}{p_{\min}^4} + \frac{2\|p_\mathcal{T} - p_\mathcal{S}\|}{p_{\min}^2}\|\psi - \hat{\psi}\|_\infty)\mathbb{E}_{x \sim \mathcal{D}_\mathcal{S}}\left[\mathbf{1}[h(x) \neq \hat{h}(x)]\right] \tag{63}$$

$$+ \|\psi - \hat{\psi}\|_\infty^2\right]D_{\chi^2}(\pi\|\mu)R_{\max}^2 \tag{64}$$

where the $\chi^2$-divergence is defined as:

$$D_{\chi^2}(\pi\|\mu) := \mathbb{E}_\mu\left[(\frac{\pi}{\mu})^2\right] \tag{65}$$

Equation (64) is due to:

$$(\gamma(x) - \hat{\gamma}(x, \hat{\psi}))^2 \tag{66}$$

$$=(\gamma(x) - \hat{\gamma}(x, \psi) + \hat{\gamma}(x, \psi) - \hat{\gamma}(x, \hat{\psi}))^2 \tag{67}$$

$$\leq(\gamma(x) - \hat{\gamma}(x, \psi))^2 + (\hat{\gamma}(x, \psi) - \hat{\gamma}(x, \hat{\psi}))^2 \tag{68}$$

$$+ 2|\gamma(x) - \hat{\gamma}(x, \psi)||\hat{\gamma}(x, \psi) - \hat{\gamma}(x, \hat{\psi})| \tag{69}$$

$$\leq(\frac{\|p_\mathcal{T} - p_\mathcal{S}\|^2}{p_{\min}^4} + \frac{2\|p_\mathcal{T} - p_\mathcal{S}\|}{p_{\min}^2}\|\psi - \hat{\psi}\|_\infty)\mathbf{1}[h(x) \neq \hat{h}(x)] \tag{70}$$

$$+ \|\psi - \hat{\psi}\|_\infty^2 \tag{71}$$

## C. Finite Sample Analysis for Reweighting Factors

The proof for our main theorem (Thm 3.7) follows directly from the Lemma C.1 and Lemma C.2.

Throughout this section, let $\sigma_{\min}(\cdot)$ denote the smallest singular value of a matrix, $\|\cdot\|_{\mathsf{op}}$ the operator norm of a multilinear operator, and $\epsilon_X := \|\hat{X} - X\|_{\mathsf{op}}$ for a matrix $X$.

### C.1. Perturbation Bound: Proof for Lemma C.1

We first state the perturbation lemma for the estimation of $\psi$:

**Lemma C.1.** *Define $\alpha$ as $\frac{\epsilon_{M_\mathcal{S}}}{\sigma_{\min}(M_\mathcal{S})}$, then we have the following bound for $\hat{\psi}$ from the joint diagnolization in Procedure 3:*

$$\|\psi - \hat{\psi}\|_\infty \leq \frac{1}{1 - \alpha}\frac{1}{\sigma_{min}(M_\mathcal{S})}(\epsilon_{M_\mathcal{T}} + (\alpha + \frac{\alpha}{\sqrt{1 - \alpha}})\|M_\mathcal{T}\|_{op}) \tag{72}$$

The proof of the lemma is similar to that of Lemma 4 of (Chaganty & Liang, 2013).

Therefore if empirical estimates of $M_\mathcal{S}$ and $M_\mathcal{T}$ are consistent and thus $\epsilon_{M_\mathcal{S}}$ and $\epsilon_{M_\mathcal{T}}$ are close to 0, the estimation error $\|\psi - \hat{\psi}\|_\infty$ goes to 0.

### C.2. Concentration bound for operator norm of the error for asymmetric moment

**Lemma C.2.** *$\forall\alpha, \beta \in \{a, b, c\}, \alpha \neq \beta$, and $\forall u \in \{\mathcal{S}, \mathcal{T}\}$. Let $\hat{M}_u^{\alpha\beta}$ be the empirical estimation of $M_u^{\alpha\beta}$ from $N$ samples for some $N \geq \frac{1}{2\epsilon^2}\log\frac{1}{\delta}$. Then with probability greater than $1 - \delta$,*

$$\|M_u^{\alpha\beta} - \hat{M}_u^{\alpha\beta}\|_{op} \leq \epsilon, \tag{73}$$

*Proof.* We use McDiarmid's inequality for some $F : \mathcal{X}^N \to \mathbb{R}$ defined as $F(A_1, \cdots, A_n) := \|\frac{1}{n}\sum_{i=1}^n A_n - \mathbb{E}[A]\|_{op}$ to prove the lemma. We have:

$$F(A_1, \cdots, A_i, \cdots, A_n) - F(A_1, \cdots, A_{i'}, \cdots, A_n) \leq \frac{1}{n}\|M_1(A_i) - M_1(A_{i'})\|_{op} \leq \frac{1}{N} \tag{74}$$

The last inequality followed from the fact that, for any domain and any superscripts $(\alpha, \beta)$, $\max\{\|\hat{M}^{\alpha\beta}\|_{op}, \|M^{\alpha\beta}\|_{op}\} \leq 1$.

By McDiarmid's inequality, we have the above concentration bound for the operator norm of the estimation error. $\square$

**C.3. Perturbation Bound for $\epsilon_M$ with errors on asymmetric statistics**

**Lemma C.3.** *For the symmetrized moment defined as*

$$\hat{M}_u = \hat{M}_u^{(cb)}(\hat{M}_u^{(ab)})^{-1}\hat{M}_u^{(ac)} \tag{75}$$

$\forall u \in \{\mathcal{S}, \mathcal{T}\}$ *We have*

$$\|M_u - \hat{M}_u\|_{op} \leq \|(M_u^{(ab)})^{-1}\|\|M_u^{(ac)}\|\epsilon_{M_u^{(cb)}} + \|\hat{M}_u^{(cb)}\|\|M_u^{(ac)}\|\epsilon_{(M_u^{(ab)})^{-1}} \tag{76}$$

$$+ \|\hat{M}_u^{(cb)}\|\|(\hat{M}_u^{(ab)})^{-1}\|\epsilon_{M_u^{(ac)}} \tag{77}$$

$$= \frac{\sigma_1(M_u^{cb})}{\sigma_{\min}(M_u^{ab})}\epsilon_{M_u^{cb}} + \frac{\sigma_1(M_u^{cb})}{\sigma_{\min}(M_u^{ab})}\frac{\sigma_1(M_u^{cb}) + \epsilon_{M_u^{cb}}}{\sigma_{\min}(M_u^{ab}) - \epsilon_{M_u^{ab}}}\epsilon_{M_u^{ab}} \tag{78}$$

$$+ \frac{\sigma_1(M_u^{cb}) + \epsilon_{M_u^{cb}}}{\sigma_{\min}(M_u^{ab}) - \epsilon_{M_u^{ab}}}\epsilon_{M_u^{ac}} \tag{79}$$

*Proof.* The proof is a simple combination of the Fact C.4 and C.5. $\qquad\square$

**C.4. Some Facts**

For any matrix A and its estimation $\hat{A}$, we have the following facts:

**Fact C.4.**

$$\|\hat{A}^{-1}\| = \frac{1}{\sigma_k(\hat{A})} \leq \frac{1}{\sigma_k(A) - \epsilon_A} \tag{80}$$

**Fact C.5.**

$$\epsilon_{A^{-1}} := \|A^{-1} - \hat{A}^{-1}\|_{op} \tag{81}$$

$$= \|A^{-1}(I - A\hat{A}^{-1})\| \tag{82}$$

$$\leq \|A^{-1}\|\|\hat{A}^{-1}\|\epsilon_A \tag{83}$$

$$= \frac{\epsilon_A}{\sigma_k(A)(\sigma_k(A) - \epsilon_A)} \tag{84}$$