# OpenReview forum: "A Spectral Method for Off-Policy Evaluation in Contextual Bandits under Distribution Shift"
_ICML.cc/2019/Workshop/RL4RealLife — Submitted to RL4RealLife 2019_

### Official Review · AnonReviewer1 · 2019-05-20
**Assumptions too strong**

**Rating:** 2
**Confidence:** 1

**Review:**

The authors tackle the problem of off-policy evaluation for contextual bandits under distribution shift; i.e., when both the policy and the distribution over contexts change.  The authors propose to model the change in context distribution via a notion of "intent."  That is, there is some latent intent associated with each context; while the distribution of context given intent is unchanged, the distribution over intents does change.  Given knowledge of the probability distributions of context given intent, the authors propose a spectral method for estimating the off-policy propensity ratios comparing probability of intents in one distribution vs. another.  Evaluation of a new policy is then straightforward.  Experiments show improvement over some baseline methods.

My comments:
-- The assumption of access to the true f_a,f_b,f_c is a very strong assumption.  This is almost always not satisfied in practice.
-- I did not understand why there are only three f_a,f_b,f_c.

I only have a basic knowledge of this area, so my confidence in rating is not strong.  Still, I think the assumption of access to the true f_a,f_b,f_c is too strong and precludes the use of this algorithm in the real world.

---

### Official Review · AnonReviewer2 · 2019-05-28
**Off-policy evaluation for contextual bandits, with specific setting of intent-based covariate-shift**

**Rating:** 3
**Confidence:** 4

**Review:**

In this work the authors studied a IS-based off-policy evaluation (OPE) method for contextual bandits, under the specific assumption that there is a categorical intent variable that governs the underlying data-generation. Instead of addressing the distribution-shift via the policy ratio, they further extend IS by taking the ration of different intents into the account. Furthermore, under the assumption of independent intent, they propose to learn the confusion matrix by the counting-based approach. They further analyze the IS estimator with this additional intent ratio. Potentially  this extension may reduce the variance of vanilla OPE (with only policy ratio). To illustrate the performance, they also ran this OPE task on MNIST classification task

In general I find the topic of this paper interesting. My comments to this paper are: 1) I am quite unsure how realistic this assumption of underlying intent is. More motivations/examples on this problem formulation are required in sec 2.2; 2) The label-independence assumption on conditional distribution seems fine, but potentially can more ideas from graphical models (such as VAEs) be used to learn the intent distributions? 3) Is it possible to compare the MSE bounds of this intent-based IS (I believe this estimator is biased) with that of vanilla IS (which is variance, because of its unbiasedness)? The current result in Theorem 4.2 is a reasonable bound, but it doesn't seem to tell me how the additional bias of the intent-based IS leads to lower variance. May be I am missing something. 4) The experiments are too simple, with too few baseline comparisons. For example the common UCI classification-to-bandit datasets would be some other alternatives benchmark to try out. What is the baseline method (green), is that vanilla importance-sampling? Seems it's not defined.

---

### Decision · Program_Chairs · 2019-05-28

Reject